# "What's the best way to document information concerning psychiatric patients? I just don't know"—A qualitative study about recording psychiatric patients notes in the era of electronic health records

Olga Chivilgina[1]*, Bernice S. Elger[1,2], Mehdi Mohamed Benichou[3], Fabrice Jotterand[1,4]

**1** Institute of Biomedical Ethics, University of Basel, Basel, Switzerland, **2** Unit of Health Law & Humanitarian Medicine at the Institute for Legal Medicine, University of Geneva, Geneva, Switzerland, **3** University Hospital of Geneva, Département de santé mentale et de psychiatrie, Service de psychiatrie de liaison et d'intervention de crise, Geneva, Switzerland, **4** Center for Bioethics and Medical Humanities, Institute for Health and Equity, Medical College of Wisconsin, Milwaukee, United States of America

* olga.chivilgina@unibas.ch

**Data Availability Statement:** Transcripts of the interviews cannot be shared publicly because data

## Abstract

This paper reports the results of a qualitative study regarding the main attitudes and concerns of Swiss psychiatrists related to the utility, usability and acceptability of EHR and how they address the pitfalls of sharing sensitive information with other parties. A total of 20 semi-structured interviews were carried out. Applied thematic analysis was used to identify themes with regard to participation. Three main themes were identified: 1) strengths of the use of EHR in the clinical context; 2) limitations of EHR; and 3) recommendations on preserving confidentiality in health records. The study shows variable practices of EHR use in psychiatric hospitals in Switzerland and a lack of standards on how to document sensitive information in EHR.

## Introduction

Electronic healthcare records (EHR) have been used in Switzerland for several years, since a federal law on patients' electronic health records (EHR) passed in 2015 and entered into force in April 2017.

The technology aims at providing effective and high-quality care, since information exchange is beneficiary for the holistic treatment of patients, and it facilitates comprehensive evaluations and treatment planning. EHR are also used in psychiatry but raise ethical concerns, as psychiatrists often deal with "very sensitive" information about their patients (Data Protection Law; [1]. Psychiatric records often contain information on patients' present symptoms, with a descriptive narrative of their life experience, including sensitive details of possible psychological trauma, domestic violence, incarceration, sexual encounters, and substance

sharing was not included in the informed consent. Since the participants have not provided consent to share their data, study materials cannot be made openly available. However, we will share portions of the study material relevant for the manuscript to those who wish to have it. Please contact our Institute for gain access to this data: a. loschnigg@unibas.ch The data available upon request can constitute the "minimal data set", which is required to replicate the reported study findings.

**Funding:** Chivilgina Olga: ESKAS-Nr : 2018.0181 Swiss Government Excellence Scholarship, https://www.sbfi.admin.ch/sbfi/de/home/bildung/stipendien/bundes-exzellenz-stipendien.html NO, The funders had no role in study design, data collection and analysis, decision to publish, or preparation of the manuscript.

**Competing interests:** The authors have declared that no competing interests exist.

abuse [2]. Concurrently, stigma in mental health is still persistent in society which makes psychiatric patients particularly vulnerable [3]. The World Health Organization (WHO) states that mental disorders affect one in four people but psychiatric conditions are still considered as "not normal", or somewhat deviant from the norm in most societies, and many people are ashamed to have mental health issues. For this reason a high proportion of psychiatric patients practise non-disclosure of their illness [4].

Furthermore, maintaining confidentiality of sensitive data is a recurrent issue in psychiatry. Even before EHR the ethical discussion surrounding disclosure of sensitive data in the paper reports to other healthcare specialists, insurances and with proxies has taken place. For example, some studies investigated the ethical concerns psychiatrists have regarding medical information confidentiality and its potential breaching. This study has shown a variety of considerations and competing priorities, such as protecting their patients from dangerous behaviour versus maintaining relationship for therapeutic process, that are used by mental health specialists when making decisions about confidentiality [5].

The concerns regarding data confidentiality were also reflected in studies on the perception of mental health professionals toward EHR in the United States. A 2010 study by Salomon et al. revealed that concerns regarding the protection of privacy and data safety impact the quality of psychiatric records in EHR [6]. The survey stated that 70% of psychiatrists at an academic medical centre clinic agreed with the following statement: "Because of concerns with confidentiality in electronic medical records, I use more "measured" (selective, discrete) wording in the medical record". Furthermore, the study revealed that 63% of clinicians were less willing to record highly confidential information and 83% disagreed with the inclusion of their own psychiatric records among routinely accessed EHR systems. Other studies have analysed the impact of EHR on the patient-doctor relationship or focused on the time spent on "desktop medicine" (interacting with the computer) as opposed to do face to face interaction with patients. These studies reveal that EHR are a barrier for good communication and affect the quality of the patient-doctor relationship [7, 8]. That being said, it should be noted that prior qualitative studies on EHR demonstrated that their use did not change the overall quality of the patient-psychiatrist relationship [9, 10]. Consequently, based on the lack of consensus on the impact of EHR on clinical practice further data collection is needed in order to have a more precise overview of their practical impact and their ethical implications in psychiatry.

EHR have been used in Switzerland for a long time but there has been no study investigating how mental health specialists deal with the ethical challenges raised by its use. To this end, the current study aimed to identify the main attitudes and concerns of Swiss psychiatrists related to the utility, usability and acceptability of EHR, and how they address the pitfalls of sharing sensitive information with other parties.

## Methods

Our research project, including the interview guide, information and consent forms, was approved by the competent cantonal research ethics committee (Ethics committee of Northwest and Central Switzerland, BASEC 2019–00040), and written informed consents were obtained from all participants. The individuals in this manuscript have given written informed consent (as outlined in PLOS consent form) to publish their opinions.

The COREQ checklist for qualitative studies was used to assure quality standards [11]. A purposive sample of mental health practitioners working in Switzerland was contacted by email or by phone. Interviewees comprised senior-level psychiatrists or clinical psychologists identified via teaching hospital websites in Switzerland and via snowballing based on the recommendations from other mental health practitioners. A semi-structured interview guide

with open-ended questions was developed taking into account the existing literature [2, 8, 10, 12–14]. The interviews took place in Switzerland from March 2019 to April 2020 and were carried out in English and in German. The interview guide with open ended questions and a vignette was sent in advance by email. All interviews were carried out by a researcher trained specifically in qualitative methods (OC), until data saturation was reached. In qualitative analysis, the appropriate sample size is reached when no additional data are being found whereby the sociologist can develop properties of the category, a point in the analytical process known as "theoretical saturation"[15, 16]. The chosen dataset was rich enough to enable us to reach theoretical saturation for the main structural elements of the theory. The interviews lasted for 35–55 min. 16 interviews were carried out at face-to-face meetings at the interviewees' workplaces, 4 were performed online via Skype. Data analysis was performed with the use of a software (F4 for transcription and MAXQDA for coding). Interviews were transcribed verbatim and German interviews were translated to English by OC. Transcriptions and translations were checked randomly by other members of the research team. To maintain the confidentiality of participants, their personal information was anonymized as well as the name of the hospital, of the cities or cantons where they work. The usual standards of qualitative analysis in medicine were applied to the continuous content analysis of the interviews [17, 18].

In this qualitative study, we reported results on answers related to electronic health records in the psychiatric setting. We aim to assess the benefits and ethical challenges of EHR and how they are perceived and handled by mental health specialists. Apart from the results concerning open-ended questions related to the experiences of participants using EHR, we also included data based on a clinical vignette. At the end of each interview, participants were asked to respond to a hypothetical case depicting a patient requesting not to report the diagnosis in his EHR (see text of the clinical vignette below).

*Patient L. is a 24 years old young men, who was just diagnosed with schizoaffective disorder. You are obliged by the hospital to write the diagnosis into a common electronic database. This information would be part of his EHR (in the hospital or EHR of a private psychiatrist). The patient is concerned that it could influence his future job perspectives or insurance benefits and asks not to save the diagnosis into the electronic health record system. What would you do?*

In order to systematize the results, we categorized the answers into different themes and sub-themes. We implemented inductive category development and systematic steps for data analysis, such as summarizing, explicating and structuring.

## Results

Emails were sent to 28 participants. 20 agreed to participate. Interviews took place in person, over the phone or over skype. Participants' characteristics included:

1. Work: 15 hospital psychiatrists (HP) (7 holding a leading position in hospital (sub)units and 8 senior psychiatrists), 4 clinical psychologists (CP), 1 private practitioner with longstanding previous hospital experience (PP).

2. Origin: 12 were from the German part (G) and 8 were from French part (F) of Switzerland.

3. Gender: 5 women and 15 men.

4. Age: 6 participants were 30–40 years old, 7 participants 40–50 y.o., 7 participants 50–60 y. o., 1 participant was older than 60.

5. Experience: 17 participants out of 20 discussed utility, usability and acceptability of EHR in the hospitals during the interview. Three interviews were also excluded.

All participants except the private practitioner were affiliated with a teaching hospital.

The results section is structured in a following way: First, positive aspects of EHR are presented. they raised multiple concerns regarding psychiatric patients including: 1) issues related to the accessibility of personal data in EHR, 2) multidisciplinary teams access to confidential data, and 3) writing reports and notes in EHR. Then, we observe psychiatry-related factors, which aggravate the complexity of the problem of data sharing in EHR: these are stigma in mental health, and moral dilemmas related to mental health specialists who decide either to preserve data confidentiality, maintaining patient trust, or to share it with other healthcare specialists. In light of these various challenges, participants suggested solutions on how to document in EHR and how to preserve data confidentiality, which we summarize in the tables (Tables 3 and 4). At the end of the section, we provide the responses of the participants on the hypothetical case, which was mentioned in the methods section.

## 1. Strengths of the use of EHR in the clinical context

Participants agreed that EHR render health care services more efficient and overcome problems associated with the lack of coordination within our health care system. Psychiatric patients are often treated by multidisciplinary teams (physicians, psychologists, family doctors, and caregivers), and use other healthcare services in hospitals. In addition, participants thought EHR help optimise treatment of the patients by having a quick access to medical information from different parties.

Overall, participants mentioned a number of different advantages: 1) accessibility of information and exchange between healthcare professionals (n = 15); 2) long-term archiving (n = 4); 3) structuring medical records (n = 1); 4) limited risk of data loss (n = 1); 5) accuracy (n = 1); and 6) readability (n = 1). See Table 1.

**Table 1. Overview of advantages of EHR in clinical work and technological advantages.**

| | |
|---|---|
| Accessibility of information and exchange between HCP for the patients' benefit | *"The information is readily available, and the doctors have duty [to share the information] towards different health care professionals that have to participate in taking care of these patients." (P6,HP,F)* |
| Readability | *"I think the quality of EHR is definitely better, you can always read what is written in the record. The other positive side is that we dictate the follow-up notes, and the secretaries write them. There is a lack of psychiatrists in Switzerland, therefore we also have doctors who are not German native speakers and the language quality of the notes improves because the secretaries can then correct them."(P19, HP,G)* |
| That the EHR system can provide a structure for a medical record | *"EHR can also help doctors to control the work processes, in which there are guidelines on how to document, e.g. also with legal forms for involuntarily confinement and its prolongation, in EHR they are designed so that they are complete." (P10,HP,G)* |
| Limited risk of data loss | *"Maybe another advantage is that the risk of data loss is limited." (P8,HP,F)* |
| Accuracy | *"Prescriptions are more accurate and the nurses see the prescriptions very quickly, so it should be beneficial for the patients." (P19,HP,F)* |
| Long-term archiving | *"The data is stored long-term. The paper can be destroyed or burned, and with EHR, data is archived permanently.." (P5,HP,G)* |

## 2. Limitations of EHR

Most participants also mentioned limitations related to EHR, especially regarding data sharing (n = 19). However, the issue was perceived differently in each hospital, as hospitals throughout Switzerland organize psychiatric files in different manners and, according to our interviewees, practices might even vary within the same hospital. Several participants reported that their hospitals have implemented a special protection system within their EHR, which allow the doctors from non-psychiatric departments to access the patient files under specific conditions, such as providing expert advice or emergency care.

> *"We have a security system, when we need to consult a patient in another department. So we have a notification that requires us to "break the glass", which means to go beyond our usual authorization access (vitre brisée). So when you want to go check the file, you have to "break the glass" to enter it and the system will record it. Therefore, you have to state the reason for which you need to consult a file that you were not initially granted access to. And the reasons must be justifiable and ethically sound. E.g. you have an expertise, an emergency, etc." (P7,HP,F)*

Furthermore, it was mentioned that some hospitals restrict access to the information within a one department, so that only doctors involved in the treatment of a particular patient can see the record.

> *"Access to the patient files is actually allowed only for those who are involved in the treatment. Theoretically, as a medical director, I have access to everything, but otherwise it [the access] is limited to people who are working on the case. We also monitor that non-authorized persons do not have access to this data." (P16,HP,G)*

Interviewees also reported that in other hospitals, psychiatry departments have technical barriers that do not allow clinicians to share patient files with other departments, despite being part of the same hospital. These limitations prevent access to the medical files even in case of an emergency.

> *"We, as a psychiatric university clinic, here in [city] practically never share our data with the somatic departments within the hospital, even during an emergency, there is no possible access.." (P10,HP,G)*

All mental health specialists mentioned that EHR enhance the risk of data abuse, breaches in confidentiality and privacy, and inappropriate record sharing (detailed description is provided in Table 2). Many participants (n = 6) have reported keeping separate notes apart from the EHR.

> *"Every time I see a patient, I enter a note in the system, stating that I have seen the patient today, and I will also note if I changed the treatment, the medication, or if I specified some condition, for example "not able to work", or if something major happened. But I keep all the patient's personal information on paper."(P6, HP,F)*

Most participants disclosed that they are often facing constraints between data sharing and data protection while writing their reports in EHR. The main reason in favour of protecting data in psychiatry is the nature of the issues discussed with psychiatrists or psychotherapists. The information is often very personal and sensitive: patients disclose to therapists their

**Table 2. Concerns.**

| | |
|---|---|
| Risk of data abuse (n = 3) | *"I see the problem of data protection and a potential for data misuse in the fact that now it is commonplace that patients are asked to give their consent for sharing their medical data with insurance companies, authorities, etc. The patients receive a flat rate confirmation that the information can finally be obtained by third parties. And it is unclear whether they were even aware of what they were signing and the depth." (P2,HP,G)* |
| Explicitness of the Informed consent (n = 4) | *"This is still a question for me, how detailed the consent for the electronic patient file should be written. I could imagine that many people would say "well, everyone can look at my somatic findings, but I would not like everyone to see my entire mental health story because that can potentially have implications." (P11,HP,G)* |
| Data protection (n = 14) | *"The most important task for the future, for electronic archiving, that we build a system, a security system. Who can read what, and do you release all data or only certain data." (P5,HP,G)* |
| Data ownership (n = 4) | *"In our files the patients don't have a control over it[the information], the patients cannot decide who will able to see it or not. and that is a real fact, that we have even us, as clinicians, who can't control it while typing the information [. . .] if the patient was here 2 weeks ago, and I saw him and I type something concerning this patient. And then if he comes back 2 weeks later and is in the emergency department for totally different reason, like abdominal pain or a headache, the emergency will still have an access to that file, and all his information on a psychiatric level." (P14,HP,F)* |
| Wrong information might be saved in a system (n = 2) | *"Sometimes, the psychiatric diagnosis isn't always right. So, to change the diagnosis, 2 or 3 years later is also very difficult. Because, if it's saved in some documents, you will read it every time again, so if in the university hospital, for example, someone gets the diagnosis schizophrenia, and you re-read the diagnosis several years after, and patient doesn't actually have schizophrenia, and that's also a problem.."(P4,HP,G)* |

internal conflicts, which refer not only to them, but sometimes also to their proxies, such as family problems and interpersonal difficulties. The sensitive nature of the information is, according to the interviewees, the major challenge in psychiatric documentation as exemplified by the following statement:

> *"Health data in each area of medicine is sensitive, but in psychiatry, these are special data, which sometimes contain a whole life story of the patient and involve a family history[. . .] And, of course, such information is very explosive data, if this kind of information fell in the wrong hands, the outcomes can be incurable." (P15,HP,G)*

Also, some intricacies mentioned by patients and the patients' interpretation of their past situation and emotional aspects are difficult to document. Subjectivity in the interpretation of data was pointed out as another particular attribute of the information communicated in mental health care. This is a particularly challenging issue in psychiatric documentation as the information has meaning for the patient and the therapist, but it may not always be objectively truthful.

> *"Sometimes there is an uncertain information, for example, when the patients say that they are not quite sure that the event actually happened in childhood. So, you do not even know if that occurred in the past, but that is now written in the record." (P5,HP,G)*

## 3. Mental illness and stigma

Participants (n = 6) highlighted that stigma is an ever-present issue within mental healthcare, and that it should be considered when information regarding psychiatric consultations is

**Table 3. Recommendations on how and what to document in EHR.**

| | |
|---|---|
| Need for standardisation (n = 2) | *"I think as long as this is not standardized, also with regard to the format of what should be in there and what can be taken out of it, it is not so useful. Otherwise it will be always a bigger and bigger file. And there is a lack of time to read through there. Then if you read in there, you have to indicate in the system where the information is updated and put together. That is the question of what and how must be documented." (P13,HP,G)* |
| Write detailed (n = 2) | *"And I decided, we have better not to write down non-objective facts. And what we feel is not an objective fact. But from the legal point of view it can be important too.[. . .] Once there was a legal issue and the judge said "from your documentation it seems that you were not in touch with the patient" and I said "no, we just don't write this." (P8,HP,F)* |
| Write less and common (n = 4) | *"But we do have to write shorter. . . In my team, we are trying to be very careful not to type the details of the trauma. If we have a patient that for instance will confide to you that we has been raped in 1970 by her father and that there was a recurrent trauma. We won't go in these detail in the medical file. We'll just talk about post trauma. . . maybe we will use the word "rape", or may be "Posttraumatic trauma" and will stick to the DSM 5 or ICD 10 Diagnosis and that's it. But we won't go into the details because of that problem that we are aware that literally any person in this hospital can have an access to this data." (P14,HP,F)* |
| Write the information useful for treatment (n = 3) | *"If the patient shares with me some sensitive information, there are 2 possibilities, either the information is personal and not useful for treatment, so you don't write it on the file, whether it's on paper or electronic file, you don't write it. And if it's useful for the treatment you have to write it down, the same in a paper file or in an electronic file.." (P18,HP,F)* |
| Document both diagnosis and medication (n = 1) | *"It is important that the diagnosis and medications are stored. If the patient is in the emergency because of an accident, the doctors and other specialists should know it, because psychotropic drugs can have a higher interaction potential and this is clearly better to know what medicine to give to the particular patient in an emergency." (P16,HP,G)* |
| Prioritize by relevance (n = 1) | *"That's the question of how things are saved in the EHR. It's useful if you have looked through the patient files or the PDFs. Nobody will have time to read the documents from the past 10 years that were scanned. So that is certainly the question of what is stored and what is relevant for information in an emergency: allergies, intolerance and certain previous illnesses are very important, so mainly medical problems.." (P5,HP, G)* |
| Avoid saving sensitive information (n = 3) | *"We try to be careful on the type of the information we type in this files. Because something can be a bit stigmatizing and. . .there are patients who give us information and trust us to keep it. . .and it may concern their past abuse and past trauma. That's very intimate information and very sensitive. And that should not be displayed there because these facts don't necessary lead to a better explanation of a medical condition.." (P17,HP,F)* |
| Save medication, and not diagnosis (n = 2) | *"We need to save the information about medication, not a diagnosis. Sometimes, the psychiatric diagnosis isn't always right. To change a diagnosis is not so easy. And, perhaps, it will come back on the central record again and again. On the other side, to save the medication it's for many doctors also important. That's why it could be an advantage of this system. So you see, a patient takes neuroleptics, but not everyone, only doctors can say, ok that is for psychosis or something like this. So there are pitfalls,it's a bit difficult but I think it's worse than the system we have now."(P4, HP,G)* |
| Save the information required by law (n = 2) | *"What's the best way to document information about psychiatric patients? I just don't know. Because it is very complex.. [. . .]So for me it is relevant that the legal documentation that is necessary is in there and a reminder that I know next time, what have I discussed with the patient, right? But I am not writing every detail of what they tell me about their life or how they are going." (P6,HP,F)* |

(*Continued*)

**Table 3.** (Continued)

| Save the information required by insurance (n = 1) | "I can write that we were struggling, fighting for money and time. We fought for liability, for guilt, for faint. I can use words like that. I also do this in the health insurance reports, I want health insurance companies to say, please, go on [with the therapy]for a one year more. So I have to bribe the health insurance company with some kind of information that the doctor who sits there and who is the internist, or lung specialist or surgeon—who is usually not a psychotherapist, the medical officer of the health insurance company, almost never. I have to explain something that maybe he understands how the treatment progresses. And I have to tell him about things that didn't actually happen. I have to tell him that the man, who has a new child with his current wife, but his two adult children from his first marriage, are not doing well. He is not here. I can say that "and the patient said "I'm never fully present". He sees that he is never here and now. And it starts hurting him. But still too little to be present. First, it is thought, there is no power in the thought yet". I can already say that to insurance. And I do that too. But I don't see any situation, emergency situation, for example, or hospitalisation, where it would help or if another psychotherapist will see the notes." (P1,PP,G) |
|---|---|

reported in EHR. According to some participants, many of their patients are ashamed of their mental health condition and hence wish to protect the information they disclosed to their mental health specialist, including the diagnosis.

*"Many patients are telling me "I'm going to say that. But please don't write it. Or please don't make any notes of it". They wanted to keep it between the two of us. I think because there is a legitimate fear." (P14,HP,F)*

Several participants (n = 4) raised concerns about mental illness-related stigma in relation to other health care professionals. Indeed, there is a concern that other health care professionals can access the EHR data which could negatively affect the quality of care psychiatric patients will receive in a non-psychiatric department.

*"We know that psychiatric patients are often poorly cared for in somatic medicine because they have psychiatric diagnosis." (P4,HP,G)*

## 4. Trust and transparency

Data sharing between health specialists is beneficial to the patient. For instance, physicians in the emergency department can make an informed decision taking into account relevant information regarding mental health disorders that may affect their patients. In many situations, for safety reason, it is important to know the patient's medication to avoid unsafe medical interactions. The more information physicians have about a patient, the more precise the diagnosis and treatment options will be. Ultimately, the process will result in higher quality of care and overall well-being of the patient.

*"I'm treating that patient, and if the patient is going to the emergency room, for the doctor there it's important to know, that this patient is treated in my service for this and this, and receives such medications." (P5,HP,G)*

The therapeutic alliance is a key component in psychiatry. Building a proper connection requires transparency from both the patient and the mental health specialist. *"In psychotherapy the relationship is the major component of the treatment. The quality of the relationship you*

**Table 4. Possible ways to preserve data confidentiality.**

| | |
|---|---|
| *Better regulation (n = 2)* | "Well, I think that EHR simply need a new clear legislation. There are still legal uncertainties in the areas of data protection and also of data exchange with other institutions, with other specialists, for further use in the new cases. There is a legal uncertainty and also technical difficulties up to now. I think we need new rights and technical methods. Otherwise, many clinicians would be interested in using this and reducing all redundancies. I think it's more a question of legislation, or maybe design, clinical guidelines and internal requirements." (P10,HP,G) |
| *Heighten Doctors' responsibility (n = 3)* | "Regarding privacy, it should be not a problem, if all the doctors and all the medical staff respect the data privacy." (P6,HP,F) |
| *Purpose related informed consent (n = 2)* | "In principle, one would have to prohibit the fact that there is a general consent for access to the file in EHR, the consent must be obtained and documented separately for each individual access by different people, and should be limited in time or limited to a certain number of accesses." (P2,HP,G) |
| *Patient participation (n = 4)* | "For me, a medical record is something personal, that belongs to the patient, something that he owns. It's not mine. I have just a copy. And we have to make an inversion between the patient and us in hospital. And the patients have to decide which data they want to share. But there is also some sensitive data or some data that makes us not proud about it. And maybe there are some personal facts that are difficult to share, as the information about psychiatric hospitalisations, or psychiatric medication. But anyway I think it would be better if the patient can decide, if he wants to blind some data on it." (P7,HP,F) |
| | "The patient himself should consider this background and make a trade-off decision, if he wants other doctors to know about this diagnosis." (P11,HP,G) |
| | "One has to involve the patients in this planning for the electronic patient archive or to include representatives of the patients. They have to join in the discussion, participate in it. Nothing about us without us. This is very important. And then a working group simply has to set up rules, pro and contra, release full information, or release it with restrictions, I can't answer that now—it's a complicated process."(P16, HP,G) |
| *Limiting access to the information in the electronic patient records (n = 7)* | "The data in psychiatry is more sensitive and that is more data, much more comprehensive and we need a system, or a rule, an algorithm that tells us how much data comes in the area that every doctor reads can, and that there may also be data that not everyone can read, only the practitioner. I could imagine, that certain sections from life history are said that are not released and that they can only be read by a narrow circle of people. But if someone is, for example, diabetic or has an allergy to penicillin, everyone must know that, of course. Maybe you have to prioritize; Data that is accessible to everyone who can read the database and data that is not accessible to everyone.."(P13,HP,G) |

have with the patient. . .you become like friends with them somehow. You have to be close, and you have to earn the full trust of the patient to help him or to help her to change his/her behaviour." (P7,CP,F)

Last but not least, therapists are often caught in dilemmas that involve preserving confidentiality, maintaining patient trust, and data sharing. This conflict was highlighted by one of the participants as follows:

*"You know, you have this inner struggle, on the medical benefits of knowing, and the ethical pitfalls of sharing, related to stigmatization. So, it's a good for a bad. I don't know what is more important." (P14,HP,F)*

## 5. Lack of clinical guidelines, lack of agreement

As highlighted by the participants, EHR technology allows for quicker access to medical files by health specialists. Unfortunately, this technology has also some limitations as it may compromise the confidentiality of sensitive data. Consequently, the data security and privacy risks influence the writing of medical documentation.

Our study has shown that the balance between data sharing and sensitive data protection remains a personal choice in the Swiss psychiatry setting. Each participant provided an individual "solution" on how to deal with the mental health information of patients. In the array of opinions, participants differ concerning the type of information and the degree of detail they considered indispensable to be recorded in their EHR. In particular, they criticise the lack of standards for documenting EHR. Many participants agreed that excessive information is unpractical. Thus, there is a need to prioritize the information according to its clinical relevance. However, even psychiatrists from the same clinic seemed to use different approaches. Common strategies chosen by several participants were "to write less and in common phrases (for example, naming the disease class "mood disorder" instead of "depression"), or to record the information strictly useful for treatment. For others, it was still important to record detailed information (See Table 3)

As mentioned above, the majority of participants perceived maintaining confidentiality of psychiatric diagnosis as the major challenge of the EHR. Participants provided several solutions for dealing with the privacy issues related to records.

## Vignette, a confrontation between obligation to record properly and patients' will

Some participants (n = 4) mentioned that they often faced the situation of patients not wanting to have their diagnosis recorded in the EHR. They stated that many patients wish to keep some information about their mental health condition private.

*"It happens from time to time that patients say, "I wouldn't like that you documented something about it"."(P2,HP,G)*

*"The problem, when the patient says, "please do not write down this diagnosis", we already have that [a lot]. I always have younger patients who are diagnosed with schizophrenia and who then sit here and say, "Dear Dr., please call it "depression", or "anxiety disorder", but just not "schizophrenia". Because it is in the files of the university hospital and then I apply for a place to study, or for a job, or go abroad, for money and funding, then somebody would read,." (P5, HP,G)*

There were two main approaches to solve this dilemma:

The majority of participants would prefer to save the diagnosis into EHR and to convince the patient to accept it. The mental health professionals interrogated mentioned that precise information will help other doctors provide a better treatment for the patient. Consequently, the patient would directly benefit from it figuring on the EHR.

*"Of course I would discuss with the patient that there are also potential advantages if other doctors know that he has this disease, because then they will be more aware, for example, of his medication. There are risks when other doctors don't know that he is taking particular drugs. Moreover, they may be able to choose a treatment for the patient more individually." (P9, CP,F)*

Another reason in favour of documenting the diagnosis was that they needed to justify the treatment to insurance companies for reimbursement purposes.

*"I have to make the diagnosis because then the health insurance company will cover the treatment." (P2 HP,G)*

Some interviewees stated it was their professional obligation to write a proper diagnosis or that they did not see an alternative to what they define as a duty of care.

*"I think I have an obligation, so I would not document it wrong. If his condition meets clearly the diagnostic criteria, I would document it and demonstrate to the patient that he has the right, that we are under confidentiality and that there is no way out of this documentation." (P10,HP,G)*

Alternatively, some participants took into account the wish of the patient and considered the use of an "umbrella" term, i.e. a more common term for the group of diseases, instead of writing the precise diagnosis.

*"If the patient does not want to disclose his diagnosis, we have to first and foremost follow the patient's desire. I would not disclose it. I will write a more broad diagnosis, because you do have to understand as well in my mind-set, and from my interaction with my colleagues, that an in-depth knowledge of diagnosis such as "schizoaffective disorder" overall is not very common. So, yeah, I think he is right to be scared of stigma. Because there is stigma, concerning this type of diagnosis. From my clinical experience. So I would just write another diagnosis, that still encompasses part of symptoms that he has. . .I mean, we're not going to lie. But we are not going to give a precise diagnosis on that." (P14,HP,F)*

*"I would tell him that I won't falsify the diagnosis, we can consider whether to formulate, for example, the entire ICD-10 diagnosis with all adjectives and such or something general, "psychosis" or "psychotic state", we can talk about it. But I do not find an essentially wrong diagnosis okay. I would try to explain this to the patient.." (P5,HP,G)*

*"Then I'm perhaps giving him some other diagnosis, something like. . .if he as a schizoaffective disorder, and you see him one time, you can give him an «brief psychosis», it normally lasts for 1 month and then it's over." (P4,HP,G)*

## Discussion

In this section, we discuss EHR in a context of the Swiss healthcare system and analyze the problems mentioned be participants. Then we outline the ethical issues associated with sharing sensitive data in the electronic patient dossier (EPD).

### Electronic hospital records

This study provides a unique perspective of variable practices of EHR use in psychiatric hospitals in Switzerland. To our knowledge it is the first qualitative study examining the perceptions of Swiss mental health specialists concerning EHR and their attitudes toward documenting mental health care in electronic medical records. As electronic documentation of mental health care raises ethical challenges worldwide, our findings are also relevant for other countries where EHR are widely implemented, such as North American and many European countries, [19, 20].

EHR have been utilized in many Swiss psychiatric hospitals, Whereas all hospitals were federally obliged to use EHR, the voluntary expanding of EHR on ambulant healthcare has evolved in different ways. In St. Gallen, Vaud and Zürich hospitals played the pivotal role in spreading EHR networks. In Geneva the patients were directly encouraged to use EHR and to have access to their data. Both approaches were shown to be successful strategies in attracting HCP in 2017. In Ticino, however, oncology private practices were reluctant to join EHR due to concerns about data sharing with patients [21] Nowadays, all hospitals and two thirds of ambulant healthcare professionals work with EHR [22], which are integrated in the overall healthcare system due to their advantages such as accessibility, accuracy and readability of records, long-term archiving, structured medical files, and limited risk of data loss.

Along with these benefits, there are still many unsolved problems in EHR implementation that deserve careful attention, such as risk of data abuse, explicitness of the informed consent, data protection, and data ownership. The issue of data protection and data confidentiality has already been discussed in other studies [23]. However, the problem of confidentiality is more prominent in light of the high emphasis on privacy in Western societies, including Switzerland. Also, many participants spoke about stigma around mental health conditions, an issue encountered in the clinical context as well as in the daily life of patients.

The struggle between ensuring the good of the patient by sharing medical information and preserving privacy and confidentiality is a recurrent problem in psychiatry. With EHR the trade-off between benefits and risks of data sharing is getting even more problematic because medical data is accessible by many parties. The quick and ubiquitous access to medical information through EHR, aimed at promoting the good of the patient, has a potential danger of stigmatization and consequently of lowering the quality of medical care. First, studies have shown that medical staff may treat psychiatric patients in a different way due to stigmatization [24–26]. Second, EHR may increase the risk of what Jones and colleagues call "diagnostic overshadowing", when physical problems are misinterpreted due the documented psychiatric state. This problem can lead to underdiagnosis and mistreatment of the physical conditions in psychiatric patients [27].

Most participants mentioned their desire to protect patients' privacy by the particular way they use EHR to document patient care. In our study, we observed a lack of consistency in documentation among healthcare providers in Switzerland which seems to be caused by the absence of guidelines specifying how to document mental health information. It is also important to mention that the EHR records may also have potential legal implications and be used in court in some cases. By law, doctors are obliged to write a correct diagnosis [28]. Only those medical procedures or assessments, as well as therapeutic discussions which were documented in the health records matter for the court. Lack of information in the record or incorrect information on a patient's history of abuse, or assault can have negative legal consequences for the attending doctor. However, this issue is beyond the scope of this paper. Our results from Table 4 and the vignette show the existence of a large variety of ways of recording information in the EHR. In order to protect patients' privacy, some participants suggested sticking to more broad information regarding the patient's mental state in the diagnosis register. In contrast, other healthcare workers provided reasons for detailed documentation of mental health concerns, such as transparency, collegiality or issues related to reimbursement by insurances.

Prioritizing medical information by clinical relevance was suggested by many of the interviewed mental health specialists. For example, there seemed to be general agreement that medication must be reported as some psychotropic drugs may cause side effects and dangerous pharmacological interactions with other medication. Information from previous treatments can be of utmost importance in predicting the patients' responsiveness to future prescriptions. Thus—although by documenting medication doctors provide direct or indirect indications

about the diagnosis underlying their prescription—the benefit of knowing about medication definitely outweigh privacy concerns. As the relevance of other information in the psychiatric record is less certain and more difficult to anticipate, interviewees varied in how they document other types of information.

Participants raised the issue concerning the impact of documentation on trust and transparency in the psychiatrist-patient relationship. From the interviews we conclude that there is no general rule recognised and followed by all psychiatrists that they should be transparent with their patients regarding what and how they document information in the medical record. Our study indicates instead that there are many individual preferences on how to document into the EHR without patients being systematically informed about these variations. This can be problematic since a lack of transparency on this can negatively impact therapeutic relationships, creating mistrust of mental health patients towards the care system and their HCP.

Moreover, EHR at the hospitals may be considered as a limiting factor for patients concerned about the privacy of their mental health condition data. These patients may seek help elsewhere, fearing stigmatization by other HCP due to their documented psychiatric condition. If patients perceive that they don't have any choice (*"I'd say to the patient: either he doesn't come to the hospital and then he will not have EHR and if he decides to come to the hospital, he has to accept that he has to have an EHR." (P19,HP,F))*, EHR can become a barrier, refraining them from consulting university hospitals and pushing them toward a certain type of private practices that still use only paper records.

To address the constraints between data sharing and data protection while using EHR in psychiatry, participants suggested a range of solutions: better regulation, heightening doctors' responsibility, obtaining a purpose-related informed consent for saving sensitive information in the record, or limiting access to the information. Some participants mentioned that the EHR content is an issue better openly discussed with the patient, which is exactly the idea behind the Electronic Patient Dossiers (EPD) initiative.

## Electronic patient dossier and future challenges

Following a new federal law adopted recently by the Swiss parliament, all hospitals in Switzerland should implement the so-called Electronic Patient Dossier (EPD) by 2021, a technology which grants patients right to assign and adapt the level of visibility of their medical information to specific health professionals, and to access and record their own data in an electronic file [29].

The EPD incorporates some of the solutions suggested by our participants. For example, it includes *"limiting access to the information in EPD"*, *"purpose related informed consent"* and *"better regulation"*, which implies the possibility to provide different levels of data access to each stakeholder.

A survey in 2020 showed that while 90% of public hospitals support implementation of EPD, only 52% of HCP in private practices would like to use EPD [30].

While adoption of EHR was voluntary for ambulatories and private practices, all certified medical professionals should adopt EPD according to the Motion 19.3955 [31], so that all ambulant care will be included in the system. The practical implementation of EPD in Switzerland was postponed due to many factors. There is a high institutional and organizational fragmentation of the Swiss healthcare system, which exists both at the institutional level, with 26 Cantons regulating the health system and a limited stance for national policies, and at the organizational levels, with many independent healthcare providers serving the same population and the strict separation between the health insurance companies and the health services providers [21, 32]. The responsibility to develop EHRs was left to Cantons or providers networks,

which resulted in the formation of different EHR systems. This process created a need of harmonization of the documenting standards as well as technical requirements of health records for EPD. Consequently, a huge administrative change is needed. The hospitals have to agree on documentation standards and to adjust their EHR systems according to the technical requirements for interinstitutional data exchange. During the transition, some private practices should use double records (hand-written and electronic). The costs for this time and effort are not yet covered by the Government [22].

Protecting sensitive data in a complex chain of multiple healthcare providers can be problematic and requires secure technologies and procedures. In order to warrant data privacy within such a complex system as EPD records all healthcare providers are supposed to be united under the so-called "circle of trust", and each healthcare provider is obliged by law to obtain conformity in terms of data privacy and data protection [33].

The solution of *"patient participation"*, as suggested by the participants in our study, is also present in the EPD. The EPD is oriented towards patients' autonomy as it makes patients the custodians of their medical records. They can subsequently grant access to their medical information to HCP. Some interviewees critically contemplated such approach, because it transfers greater responsibility to patients with mental health disorders, including those with altered decision-making capacities (during moments where capacity is present) and their representatives. The EPD is sometimes perceived as problematic for mental health care as patients can restrict access to some of their information to health specialists, without necessarily understanding its medical importance. In that case, they are held accountable for the consequences of their choices. As one participant reported (P13,HP,G), it is "good that the patient can determine who has what information" because that" implies that the patient then has more responsibility". The risk is however, that if "a wrong decision happens based on the missing information, and let's assume the patient has also considered something and thereby the decision is made that was not the right one" the responsibility transferred to the patient might be inappropriate and cause harm as the patient is "a medical layperson" and cannot determine what is important and will cause benefit or harm in the same way as a doctor can. For HCP it is often difficult to understand why patients would not want to share important information with doctors. HCP might underestimate risks to privacy and feel upset ("What is so important that the doctor can / should not know" (P13,HP,G)) thinking that patients accuse them in a non-justified way.

Based on the idea of an overriding concept of beneficence, the federal law on the EPD stipulates that in medical emergency situations healthcare professionals can get access to the medical data of the patient, even if the patient restricted access rights [29]. The EPD has thus been conceived as a possible way to solve the ethical conundrums posed by the EHR, i.e. by regulating data sharing on different levels. Nevertheless, it might create another set of issues such as a greater concern on confidentiality due to cross-institutional data exchange, and increased responsibility on the part of patients for sharing their data with other healthcare specialists.

The development of EPD and EHR also raises the question of dealing with a temporary alteration in the capacity to consent and decision making that is typical for numerous mental health conditions, e.g., schizophrenia, major depressive disorder with psychotic features, manic disorder with or without psychotic features, etc. In such situations, if patients have greater access to their medical data, they may–in a moment of decreased capacity–disclose sensitive information on social media, i.e. act in a way they would not condone when fully capable.

The latest "shocking" hijack of psychotherapy records in Finland is yet another example of events that make patients and mental health care professionals alike nervous about thorough documentation in any electronic form [34]. With the increase of ransomware and data

breaches, the relevant question of cybersecurity and thorough data encryption must also be taken into account in acquiring the patients trust into any kind of electronic health data documentation, including the EPD and EHR.

Since the sensitive information can be shared with many stakeholders in EPD, the discussion on how to document sensitive psychiatric data is going to gain even more significance for standardization of health records.

## Limitations

This research is subject to several limitations. First, we have interviewed participants in the French and German parts of Switzerland, so the results may be not generalizable to the whole country (which also includes an Italian speaking part). Second, the responses of participants could contain bias of social desirability which could mean that that practices deemed problematic were underreported. However, the fact that we found a high variability of attitudes, some more problematic than others, is an indicator that we were able to picture many facets of current practice, and that the bias of social desirability remained low. The study drew on data that were gathered at a point in time in a rapidly evolving landscape of development of healthcare IT technologies and we cannot guarantee that participants' attitudes might not have changed since the interviews took place. Finally, we have reported only on a subsection of topics addressed in our interview guide. The guide was not focused specifically on EHR, but also included structured questions on EPD and mobile apps for psychiatry. However, the responses indicate data saturation on the main themes that we report here.

## Conclusion

The digitalization of psychiatric records is a source of many controversial ethical issues as exemplified in the use of EHR and EPD since the information in these records is particularly sensitive. It is important to recognize and discuss them openly and our study has the merit to make this discussion better visible.

Our research revealed a panoply of competing opinions of Swiss mental health specialists on how to document in EHR. This reflects the various EHR systems in the fragmented healthcare system in Switzerland. There is a need to reflect further on these practices. It is important to ensure that health professionals are trained in the process of ethical decision-making involved in dealing with sensitive information in the medical record. Furthermore, ethical standards for the documentation of mental health conditions in EHR and EPD should be openly discussed and there is a need for more explicit professional guidelines. Patients' participation seems to be a very promising approach for a patient-centred care paradigm in psychiatry. Nevertheless, it imposes greater responsibility for patients who can actively manage and share their health information in cross-sectoral care and this could be problematic for the vulnerable group of patients needing mental health care. Sensitivity to the needs of this vulnerable group of people needs to be heightened when creating the future regulations on EHR and EPD. It will also be important to provide training for the mental health patients on using EHR and EPD and on how to manage confidentiality risks in an efficient and transparent way.

The recent COVID pandemic showed the importance of quick access and interoperability of data for clinical care and research, therefore EHR and EPD may positively affect healthcare efficiency. Introduction and harmonization of EHR and EPD in decentralized health systems can be problematic. Whereas some countries, such as Austria or US [35, 36], provide financial incentives to accelerate EHR adoption, currently in Switzerland there exist no such support mechanisms.

The need to protect patients' sensitive information is of great importance in the Swiss healthcare system. The discussion on what and how to document sensitive data in EHR and EPD requires the right legal and technical framework to harmonize the fragmented health care system. This experience may be of use for decision-makers in the other countries.

## Author Contributions

**Conceptualization:** Olga Chivilgina, Bernice S. Elger, Mehdi Mohamed Benichou, Fabrice Jotterand.

**Data curation:** Olga Chivilgina, Bernice S. Elger, Fabrice Jotterand.

**Formal analysis:** Olga Chivilgina.

**Funding acquisition:** Olga Chivilgina.

**Investigation:** Olga Chivilgina.

**Methodology:** Olga Chivilgina.

**Project administration:** Olga Chivilgina.

**Supervision:** Bernice S. Elger, Fabrice Jotterand.

**Writing – original draft:** Olga Chivilgina, Mehdi Mohamed Benichou.

**Writing – review & editing:** Olga Chivilgina, Bernice S. Elger, Fabrice Jotterand.

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
