## [Decision Letter · Decision Letter 0]

13 Aug 2021

PONE-D-21-12855

“What’s the best way to document information concerning psychiatric patients? I just don't know” – a qualitative study about recording psychiatric patients notes in the era of electronic health records

PLOS ONE

Dear Dr. Chivilgina,

Thank you for submitting your manuscript to PLOS ONE. After careful consideration, we feel that it has merit but does not fully meet PLOS ONE’s publication criteria as it currently stands. Therefore, we invite you to submit a revised version of the manuscript that addresses the points raised during the review process.

The reviewers' aims are to be constructive and improve the paper. Do note that both reviewers ask for more details about the policy and practice context in Switzerland. They also raise concerns about your methods which you should address. 

We look forward to receiving your revised manuscript.

Kind regards,

Maria Berghs, PhD

Academic Editor

PLOS ONE

Journal Requirements:

“NO”

Additional Editor Comments (if provided):

Both reviewers raise issues about the changed policy and practice context in Switzerland which is important to address. There are also some methodological concerns which require clarification.

Reviewers' comments:

Reviewer's Responses to Questions

**Comments to the Author**

1. Is the manuscript technically sound, and do the data support the conclusions?

Reviewer #1: Yes

Reviewer #2: Partly

2. Has the statistical analysis been performed appropriately and rigorously? 

Reviewer #1: N/A

Reviewer #2: No

3. Have the authors made all data underlying the findings in their manuscript fully available?

Reviewer #1: No

Reviewer #2: No

4. Is the manuscript presented in an intelligible fashion and written in standard English?

Reviewer #1: Yes

Reviewer #2: Yes

5. Review Comments to the Author

Reviewer #1: This manuscript provides a qualitative analysis of the Electronic Healthcare Record system used in Switzerland. The article is of great interest, and my comments are intended to improve the manuscript. Overall, the qualitative approach is novel, and provides a new perspective on the use of EHR; but more work is required to improve the narrative and framing of the results. My comments are listed below:

1. The abstract requires significant modification, it is too long and includes citations. It should be a short reflection on the rationale for the study, the aim, the method utilised and the main findings.

2. The introduction focuses on the introduction of EHR; however, it fails to provide any detail on how EHR are formed and managed in Switzerland. Does the National Government specify how EHR are organised, or is it at a regional or local level? As a reader, understanding ‘what’ the EHR is in Switzerland helps frame your findings. Another interesting element would be to provide some statistics on its use across Switzerland.

3. The authors state that EHR have been used in Switzerland for a ‘long time’ (be helpful to provide a timeframe); therefore, it is appropriate to use the study framing as ‘EHR implementation’? The study does not reflect on how EHR was implemented, but on its utility, usability and acceptability. I would encourage the authors to consider this point. If they still believe implementation is the correct phrasing, a definition would be useful.

4. The methods section is brief considering the study scope, it would be useful to expand more on the theoretical underpinning of the qualitative analysis. Further, how did the authors determine data saturation? Was a continuous analytical approach undertaken?

5. I have no criticism of the results. Thought-out and detailed.

6. The main limitation of the paper is its lack of ambition in synthesis in the discussion. It would be useful to describe how the EHR in Switzerland compares with other Nation’s EHR systems. Can any lessons be learnt from other EHR systems? And in return, what can other EHR systems adopt from the findings of this study?

Reviewer #2: This manuscript details a qualitative analysis of interviews with Swiss psychiatrists regarding several issues surrounding electronic health records. The authors faithfully report interview excerpts and do a good job on the whole of categorizing recurring themes and attitudes towards psychiatric EHR use. Below are comments that I feel would improve the manuscript:

1. The text is well-written, but I feel the organization could be clarified and improved. For example the end of the results section states "participants suggested solutions listed below", but is then followed by several pages that are mostly psychiatrist concerns/challenges (until Table 3. Recommendations). To my reading, the sections 1-5 did not follow a common theme and should be reorganized, potentially with more descriptive headings or sub-headings.

2. Some of the headings also did not make sense to me (like "3. Personal-self (patient)").

3. To my reading, some of the issues treated by the authors as distinct seemed to overlap. For example, in the Vignette: the issue of "saving diagnoses in EHR for improving patient care by sharing amounts providers" and "medical collegiality" seem almost equivalent.

4. Given the statement that all Swiss hospitals should implement EPD by 2021 (we are currently more than halfway through 2021), I think the authors should spend more time discussing whether their interviews and findings are still relevant given this new system that appears to solve many existing challenges. It would have been nice to discuss EPD in the interviews since it appears to be a large paradigm shift.

5. Some minor typos throughout: at least one instance of "HER" (should be "EHR)", "hack of psychotherapy records" in final paragraph before the Limitations (should be "lack"), capitalizing "Skype" in results section, etc.

6. The authors should check for consistency in their methods and be as detailed as possible. In the first paragraph of the Methods section, they say 16 interviews were carried out face-to-face and 4 interviews were performed online via Skype. In the Results section, they state that the 20 interviews took place in person, "over the phone", or over Skype.

7. The paper is mostly insertions of things the psychiatrists said during the interviews. In "1. Strengths of the use of EHR...", I appreciated the summarization of how many interviewees mentioned certain themes - I think the authors should repeat this for the other results sections.

6. PLOS authors have the option to publish the peer review history of their article (what does this mean?). If published, this will include your full peer review and any attached files.

Reviewer #1: No

Reviewer #2: No

---

## [Author Response · Author response to Decision Letter 0]

16 Oct 2021

We would like to thank the editor and the reviewers for careful and thorough reading of this manuscript and for the thoughtful comments and constructive suggestions, which gave us other chance to improve the quality of this manuscript. 

Responses to the reviewer 1.

Reviewer #1: This manuscript provides a qualitative analysis of the Electronic Healthcare Record system used in Switzerland. The article is of great interest, and my comments are intended to improve the manuscript. Overall, the qualitative approach is novel, and provides a new perspective on the use of EHR; but more work is required to improve the narrative and framing of the results. My comments are listed below:

Comment 1: The abstract requires significant modification, it is too long and includes citations. It should be a short reflection on the rationale for the study, the aim, the method utilised and the main findings.

Response: We absolutely agree with your critics. So, we added an abstract, as following: 

“This paper reports the results of a qualitative study regarding the main attitudes and concerns of Swiss psychiatrists related to the utility, usability and acceptability of EHR and how they address the pitfalls of sharing sensitive information with other parties. A total of 20 semi-structured interviews were carried out. Applied thematic analysis was used to identify themes with regard to participation. Three main themes were identified: 1) strengths of the use of EHR in the clinical context; 2) limitations of EHR; and 3) recommendations on preserving confidentiality in health records. The study shows variable practices of EHR use in psychiatric hospitals in Switzerland and a lack of standards on how to document sensitive information in EHR.”

Comment 2: The introduction focuses on the introduction of EHR; however, it fails to provide any detail on how EHR are formed and managed in Switzerland. Does the National Government specify how EHR are organised, or is it at a regional or local level? As a reader, understanding ‘what’ the EHR is in Switzerland helps frame your findings. Another interesting element would be to provide some statistics on its use across Switzerland.

Response: That’s a very important point. Of course, we added the information on how the EHR is organized. Unfortunately, there is lack of data on EHR use across Switzerland. We added the following paragraphs:

“EHR have been utilized in many Swiss psychiatric hospitals, Whereas all hospitals were federally obliged to use EHR, the voluntary expanding of EHR on ambulant healthcare has evolved in different ways. In St. Gallen, Vaud and Zürich hospitals played the pivotal role in spreading EHR networks. In Geneva the patients were directly encouraged to use EHR and to have access to their data. Both approaches were shown to be successful strategies in attracting HCP in 2017. In Ticino, however, oncology private practices were reluctant to join EHR due to concerns about data sharing with patients (De Pietro and Francetic 2018) Nowadays, all hospitals and two thirds of ambulant healthcare professionals work with EHR (V. Pfeiffer 2021), which are integrated in the overall healthcare system due to their advantages such as accessibility, accuracy and readability of records, long-term archiving, structured medical files, and limited risk of data loss.”

 “The responsibility to develop EHRs was left to Cantons or providers networks, which resulted in the formation of different EHR systems..” 

Comment 3: The authors state that EHR have been used in Switzerland for a ‘long time’ (be helpful to provide a timeframe); therefore, it is appropriate to use the study framing as ‘EHR implementation’? The study does not reflect on how EHR was implemented, but on its utility, usability and acceptability. I would encourage the authors to consider this point. If they still believe implementation is the correct phrasing, a definition would be useful.

Response: Thanks for your recommendation. We have specified a timeframe and changed the phrasing. 

Comment 4. The methods section is brief considering the study scope, it would be useful to expand more on the theoretical underpinning of the qualitative analysis. Further, how did the authors determine data saturation? Was a continuous analytical approach undertaken?

Response: In qualitative analysis, the appropriate sample size is reached when no additional data are being found whereby the sociologist can develop properties of the category, a point in the analytical process known as ‘‘theoretical saturation’’(Dey 2012, Glaser and Strauss 2017). The chosen dataset was rich enough to enable us to reach theoretical saturation for the main structural elements of the theory. The analytical approach was performed continuously.

Comment 5. I have no criticism of the results. Thought-out and detailed.

Response: We appreciate the positive feedback from the rewiever. 

Comment 6. The main limitation of the paper is its lack of ambition in synthesis in the discussion. It would be useful to describe how the EHR in Switzerland compares with other Nation’s EHR systems. Can any lessons be learnt from other EHR systems? And in return, what can other EHR systems adopt from the findings of this study?

Response: We added following paragraph to the discussion. 

“Introduction and harmonization of EHR and EPD in decentralized health systems can be problematic. Whereas some countries, such as Austria or US (aerzteblatt.de 26. Februar 2021, Marcotte, Seidman et al. 2012), provide financial incentives to accelerate EHR adoption, currently in Switzerland there exist no such support mechanisms.”

. 

“The need to protect patients’ sensitive information is of great importance in the Swiss healthcare system. The discussion on what and how to document sensitive data in EHR and EPD requires the right legal and technical framework to harmonize the fragmented health care system. This experience may be of use for decision-makers in the other countries.”

Reviewer #2: This manuscript details a qualitative analysis of interviews with Swiss psychiatrists regarding several issues surrounding electronic health records. The authors faithfully report interview excerpts and do a good job on the whole of categorizing recurring themes and attitudes towards psychiatric EHR use. Below are comments that I feel would improve the manuscript:

1. The text is well-written, but I feel the organization could be clarified and improved. For example the end of the results section states "participants suggested solutions listed below", but is then followed by several pages that are mostly psychiatrist concerns/challenges (until Table 3. Recommendations). To my reading, the sections 1-5 did not follow a common theme and should be reorganized, potentially with more descriptive headings or sub-headings.

Comment 2: Some of the headings also did not make sense to me (like "3. Personal-self (patient)").

Response: We deleted the sub-heading (“3. Personal-self (patient)”).

Comment 3: To my reading, some of the issues treated by the authors as distinct seemed to overlap. For example, in the Vignette: the issue of "saving diagnoses in EHR for improving patient care by sharing amounts providers" and "medical collegiality" seem almost equivalent.

Response: We deleted the paragraph to avoid repeating. 

Comment 4: Given the statement that all Swiss hospitals should implement EPD by 2021 (we are currently more than halfway through 2021), I think the authors should spend more time discussing whether their interviews and findings are still relevant given this new system that appears to solve many existing challenges. It would have been nice to discuss EPD in the interviews since it appears to be a large paradigm shift.

Response: Indeed, EPD is a large paradigm shift, however, our research is getting even more relevant in the light of data sharing with many stakeholders and consequently increased privacy concerns within EPD. So, we added the following paragraph in the discussion: 

“Since the sensitive information can be shared with many stakeholders in EPD, the discussion on how to document sensitive psychiatric data is going to gain even more significance for standardization of health records.”

Comment 5: Some minor typos throughout: at least one instance of "HER" (should be "EHR)", "hack of psychotherapy records" in final paragraph before the Limitations (should be "lack"), capitalizing "Skype" in results section, etc.

Response: As suggested by the associate editor, we have improved the typos mentioned above. We changed the word “hack” for “hijack” in order to avoid misunderstanding. 

Comment 6: The authors should check for consistency in their methods and be as detailed as possible. In the first paragraph of the Methods section, they say 16 interviews were carried out face-to-face and 4 interviews were performed online via Skype. In the Results section, they state that the 20 interviews took place in person, "over the phone", or over Skype.

Response: We improved the statement. The invitations were performed partly on phone. All interviews were carried out in person or via Skype. 

Comment 7: The paper is mostly insertions of things the psychiatrists said during the interviews. In "1. Strengths of the use of EHR...", I appreciated the summarization of how many interviewees mentioned certain themes - I think the authors should repeat this for the other results sections.

Response: We added the information on the numbers of the interviewees in the tables.

CLOSING COMMENTS TO THE EDITOR: Again, we appreciate the opportunity to revise our work for consideration for publication in the Plos One: Mental Health and Psychiatry. We hope our revision meet your approval. 

The Authors

---

## [Decision Letter · Decision Letter 1]

19 Nov 2021

PONE-D-21-12855R1“What’s the best way to document information concerning psychiatric patients? I just don't know” – a qualitative study about recording psychiatric patients notes in the era of electronic health recordsPLOS ONE

Dear Dr. Chivilgina,

Thank you for submitting your manuscript to PLOS ONE. After careful consideration, we feel that it has merit but does not fully meet PLOS ONE’s publication criteria as it currently stands. Therefore, we invite you to submit a revised version of the manuscript that addresses the points raised during the review process.

We look forward to receiving your revised manuscript.

Kind regards,

Maria Berghs, PhD

Academic Editor

PLOS ONE

Journal Requirements:

Additional Editor Comments (if provided):

The article is not of publishable standard. This has nothing to do with the data nor methods but the presentation of the findings and formatting. So, I hope you will attend to these issues. One reviewer has advised to accept but before we can accept it, I would like you to ensure that, for example, when presenting your findings that you signpost to the reader what these are.. This is why the headings did not make sense for the second reviewer. who requested major revisions. So, I would insert a short paragraph before the presentation of the findings, signposting the main themes (your headings) and explain that you will discuss these in turn. The results section with table seems to correspond to your methods (how many people participated in your research/who/sample) and the findings are about your themes that you found in your data analysis. The way it is currently laid out is confusing to the reader.

I would also encourage you to look at some other papers in the journal and see what the conventions are when sharing quotations and how they signpost. I would advise you to signpost more and also present quotations likewise so that it is the voices of your participants that are highlighted: For example, here the voice is first: 

“Many patients are telling me "I'm going to say that. But please don't write it. Or please don't make any notes of it". They wanted to keep it between the two of us. I think because there is a legitimate fear” (P14, HP,F)

At the moment , in places, your quotations are lists rather than ethical analysis of what people are saying. Could you use fewer quotations or format so we can see them clearer? The section on Trust and Transparency is good.

Again, before your discussion it might be useful to signpost to the reader what the main points will be that you are discussing.

Please check your first reference and that you do not have to mention an author?

The conflicts of interests section goes at the end of an article and not before the conclusion.

Reviewers' comments:

Reviewer's Responses to Questions

**Comments to the Author**

1. If the authors have adequately addressed your comments raised in a previous round of review and you feel that this manuscript is now acceptable for publication, you may indicate that here to bypass the “Comments to the Author” section, enter your conflict of interest statement in the “Confidential to Editor” section, and submit your "Accept" recommendation.

Reviewer #2: All comments have been addressed

2. Is the manuscript technically sound, and do the data support the conclusions?

Reviewer #2: Partly

3. Has the statistical analysis been performed appropriately and rigorously? 

Reviewer #2: N/A

4. Have the authors made all data underlying the findings in their manuscript fully available?

Reviewer #2: No

5. Is the manuscript presented in an intelligible fashion and written in standard English?

Reviewer #2: (No Response)

6. Review Comments to the Author

Reviewer #2: The authors have satisfactorily addressed all comments. I now recommend this manuscript for publication.

7. PLOS authors have the option to publish the peer review history of their article (what does this mean?). If published, this will include your full peer review and any attached files.

Reviewer #2: No

---

## [Author Response · Author response to Decision Letter 1]

24 Jan 2022

(attached as a separate document)

---

## [Editor Report · Decision Letter 2]

8 Feb 2022

“What’s the best way to document information concerning psychiatric patients? I just don't know” – a qualitative study about recording psychiatric patients notes in the era of electronic health records

PONE-D-21-12855R2

Dear Dr. Chivilgina,

We’re pleased to inform you that your manuscript has been judged scientifically suitable for publication and will be formally accepted for publication once it meets all outstanding technical requirements.

Kind regards,

Maria Berghs, PhD

Academic Editor

PLOS ONE

Additional Editor Comments (optional):

Thank-you for your hard work and patience with this article.
---

## [Editor Report · Acceptance letter]

14 Feb 2022

PONE-D-21-12855R2 

“What’s the best way to document information concerning psychiatric patients? I just don't know” – a qualitative study about recording psychiatric patients notes in the era of electronic health records 

Dear Dr. Chivilgina:

I'm pleased to inform you that your manuscript has been deemed suitable for publication in PLOS ONE. Congratulations! Your manuscript is now with our production department. 

Kind regards, 

on behalf of

Dr. Maria Berghs 

Academic Editor

PLOS ONE